# IMPLICIT VARIATIONAL REJECTION SAMPLING

## ABSTRACT

Variational Inference (VI) is a cornerstone technique in Bayesian machine learning, employed to approximate complex posterior distributions. However, traditional VI methods often rely on mean-field assumptions, which may inadequately capture the true posterior's complexity. To address this limitation, recent advancements have utilized neural networks to model implicit distributions, thereby offering increased flexibility. Despite this, the practical constraints of neural network architectures can still result in inaccuracies in posterior approximations. In this work, we introduce a novel method called Implicit Variational Rejection Sampling (IVRS), which integrates implicit distributions with rejection sampling to enhance the approximation of the posterior distribution. Our method employs neural networks to construct implicit proposal distributions and utilizes rejection sampling with a meticulously designed acceptance probability function. A discriminator network is employed to estimate the density ratio between the implicit proposal and the true posterior, thereby refining the approximation. We propose the Implicit Resampling Evidence Lower Bound (IR-ELBO) as a metric to characterize the quality of the resampled distribution, enabling the derivation of a tighter variational lower bound. Experimental results demonstrate that our method outperforms traditional variational inference techniques in terms of both accuracy and efficiency, leading to significant improvements in inference performance. This work not only showcases the effective combination of implicit distributions and rejection sampling but also offers a novel perspective and methodology for advancing variational inference.

## 1 INTRODUCTION

Variational Inference (VI) has emerged as a fundamental technique in Bayesian machine learning for approximating complex posterior distributions (Hoffman et al., 2013). Traditional VI methods frequently rely on a mean-field assumption (Blei et al., 2017), which trades off posterior expressiveness for computational tractability. To address this limitation, implicit distributions, which are typically modeled using neural networks, have been proposed to leverage their flexibility when approximating complex posterior distributions (Mescheder et al., 2017; Huszár, 2017; Titsias & Ruiz, 2019; Shi et al., 2017). Despite this added flexibility, neural networks still exhibit shortcomings in certain scenarios. For instance, the practical application of neural networks is constrained by the number of layers and neurons, and their structural design (e.g., activation functions) is often guided by empirical knowledge (Hornik et al., 1989; Krizhevsky et al., 2012; LeCun et al., 2015). As a result, the posterior approximation ability of neural networks can vary, leading to potential bias between the implicit distribution and the true posterior.

To address this issue, we propose a method called Implicit Variational Rejection Sampling (IVRS), which leverages rejection sampling (Gilks & Wild, 1992) to better exploit the strengths of implicit distributions in achieving accurate posterior approximations. We first use neural networks to construct implicit distributions that serve as proposal distributions. We then design an acceptance probability function, which is related to the density ratio between the proposal distribution and the true posterior, and apply rejection sampling to generate resampled samples. A discriminator network is used to approximate the density ratio, thereby refining the proposal distribution into a more accurate posterior approximation. By incorporating adversarial training techniques, this approach enables us to construct an Implicitly Resampled Evidence Lower Bound (IR-ELBO). We discuss the advantageous properties of the resampled distribution, particularly its reduced KL divergence with the true posterior. Extensive experiments demonstrate that IVRS outperforms traditional variational inference methods in both accuracy and efficiency.

We summarize the contributions of our paper below:

- We introduce Implicit Variational Rejection Sampling (IVRS) to combine implicit distributions with rejection sampling for improved variational inference.
- We design an acceptance probability function based on the density ratio between the proposal distribution and the true posterior, and apply rejection sampling to refine the proposal into a more accurate posterior approximation via a discriminator network.
- By incorporating adversarial training techniques, we construct an Implicit Resampling Evidence Lower Bound (IR-ELBO) and analyze the favorable properties of the resampled distribution, particularly its reduced KL divergence from the true posterior.
- We demonstrate through extensive experiments that IVRS can outperform traditional variational inference methods in terms of both accuracy and efficiency.

## 2 MODEL FRAMEWORK

### 2.1 BAYESIAN GENERATIVE MODELS

We consider an unsupervised generative model for a dataset $\mathcal{D} = \{\mathbf{x}_i\}_{i=1}^{N}$ that has latent variables $\mathbf{z}$ and model parameters $\theta$. The joint distribution of the model has the form

$$p(\mathbf{x}, \mathbf{z}|\theta) = p(\mathbf{z})p(\mathbf{x}|\mathbf{z}, \theta) \tag{1}$$

where $p(\mathbf{z})$ is the prior distribution of $\mathbf{z}$, and $p(\mathbf{x}|\mathbf{z}; \theta)$ is a parametric generative model. In traditional Bayesian models such as Gaussian Mixture Models (GMMs), this distribution is often specified through manual design. With the advancement of deep learning, particularly in generative models, this distribution is frequently parameterized using neural networks, as exemplified by Variational Autoencoders (VAEs) and similar architectures. While this framework can be extended to supervised learning scenarios, we present and illustrate our result on unsupervised learning problems.

### 2.2 VARIATIONAL INFERENCE

The goal of inference in probabilistic unsupervised learning is to model the posterior distribution of the latent variables in Equation (1). For non-conjugate models, such as those involving deep learning architectures, the posterior distribution can be highly complex and require approximate inference methods. A commonly used method is Variational Inference (VI). In VI, the goal is to approximate the posterior distribution $p(\mathbf{z}|\mathbf{x})$ with a predefined variational distribution $q(\mathbf{z}|\phi)$ according to the KL divergence. This is achieved by maximizing the Evidence Lower Bound (ELBO),

$$\mathcal{L}(\mathbf{x}, \theta, \phi) = \mathbb{E}_{q(\mathbf{z}|\phi)} \left[ \log p(\mathbf{x}, \mathbf{z}|\theta) - \log q(\mathbf{z}|\phi) \right]. \tag{2}$$

The traditional mean-field approximation assumes a factorized form for the variational distribution,

$$q(\mathbf{z}|\mathbf{x}, \phi) = \prod_{i=1}^{m} q(\mathbf{z}_i|\mathbf{x}|\phi_i), \tag{3}$$

where $m$ represents the number of factors in the decomposition, and $q(\mathbf{z}_i|\mathbf{x}, \phi_i)$ is often an analytically tractable distribution, such as a Gaussian. However, due to the often complex nature of non-conjugate posterior distributions, this restrictive assumption can lead to biased estimates.

To address these limitations, implicit variational inference approaches employ neural networks to model the variational distribution. These methods aim to capture more complex posterior structures by leveraging the expressive power of neural networks, thereby improving accuracy,

$$\mathbf{z} = f_\phi(\mathbf{x}, \epsilon) \sim q_\phi(\mathbf{z}|\mathbf{x}), \quad \epsilon \sim p(\epsilon) \tag{4}$$

Here, $\phi$ represents the parameters of a neural network, while $\epsilon$ is drawn from a simple distribution such as a Gaussian. This setup allows for more flexible posterior approximations. Recently, various algorithms have been proposed to effectively train such models, including Adversarial Variational Bayes (Mescheder et al., 2017) and Semi-implicit Variational Inference (Yin & Zhou, 2018).

Despite this flexibility, the layers and neurons in traditional neural networks have practical limitations and their structural design is often based on empirical considerations. Consequently, the

approximation ability of neural networks can vary empirically. We next present a method to mitigate these problems by incorporating rejection sampling. Although there is substantial prior research on rejection sampling, to our knowledge this paper is the first to apply this statiscial technique specifically to implicit posterior distribution modeling for variational inference.

## 3 PROPOSED METHOD

### 3.1 REJECTION SAMPLING

Rejection sampling is a standard technique to create samples from a target distribution through samples generated from a proposal distribution. Given a target distribution $p_{\text{tar}}(\mathbf{z})$ and a proposal distribution $q_{\text{pro}}(\mathbf{z})$, rejection sampling accepts a sample $\mathbf{z} \sim q_{\text{pro}}(\mathbf{z})$ with probability defined by an acceptance probability function $a(\mathbf{z})$ that is proportional to the ratio of the target density to the proposal density,

$$a(\mathbf{z}) = \frac{p_{\text{tar}}(\mathbf{z})}{M q_{\text{pro}}(\mathbf{z})}, \tag{5}$$

where $M \in \mathbb{R}^+$ and the choice of $M$ ensures that the acceptance probability is less than or equal to 1. In our model, the target distribution $p_{\text{tar}}(\mathbf{z}) = p_\theta(\mathbf{z}|\mathbf{x})$, being the true posterior of the latent variables of model structure (1). We define the proposal distribution as the implicit distribution in Equation (5), $q_{\text{pro}}(\mathbf{z}) = q_\phi(\mathbf{z}|\mathbf{x})$. Consequently, we express the acceptance rate function as $a(\mathbf{z}; \mathbf{x}, \theta, \phi)$.

However, unlike traditional rejection sampling, calculating the acceptance rate function $a(\mathbf{z}; \mathbf{x}, \theta, \phi)$ in our model is not analytical. This introduces two primary challenges:

    i) The target distribution $p_\theta(\mathbf{z}|\mathbf{x})$ is the true posterior, typically only representable in the unnormalized joint likelihood form of Bayes' rule in Equation (1).

    ii) The proposal distribution $q_\phi(\mathbf{z}|\mathbf{x})$ is an implicit distribution, often generated by through a structure that makes it difficult to determine its probability density function.

We next turn to our proposal for addressing these two issues.

### 3.2 CONSTRUCTION OF THE ACCEPTANCE PROBABILITY FUNCTION

To address the two challenges mentioned above, we start by expressing the target distribution $p_\theta(\mathbf{z}|\mathbf{x})$ using Bayes' rule,

$$p_\theta(\mathbf{z}|\mathbf{x}) = p_\theta(\mathbf{x}|\mathbf{z})p(\mathbf{z})/p_\theta(\mathbf{x}) \tag{6}$$

where $p_\theta(\mathbf{x}|\mathbf{z})$ is the likelihood, $p(\mathbf{z})$ is the prior, and $p_\theta(\mathbf{x}) = \int p_\theta(\mathbf{x}|\mathbf{z})p(\mathbf{z})\,d\mathbf{z}$ is the evidence. To ensure the acceptance probability is within the range $[0, 1]$, we can ignore the evidence term provided we choose an appropriate scaling factor $M$ such that

$$a(\mathbf{z}; \mathbf{x}, \theta, \phi) = \frac{p_\theta(\mathbf{x}|\mathbf{z})p(\mathbf{z})}{M q_\phi(\mathbf{z}|\mathbf{x})} \leq 1 \tag{7}$$

In practice, to ensure that the acceptance rate is less than or equal to 1, the acceptance rate function is usually constructed as

$$a(\mathbf{z}; \mathbf{x}, \theta, \phi) = \min\left[\frac{p_\theta(\mathbf{x}|\mathbf{z})p(\mathbf{z})}{M q_\phi(\mathbf{z}|\mathbf{x})}, 1\right]. \tag{8}$$

Due to the presence of the min function in Equation 8, gradient-based optimization can be challenging. To address this, we adopt a fully differentiable approximation inspired by Grover et al. (2018),

$$a(\mathbf{z}; \mathbf{x}, \theta, \phi) = \frac{1}{1 + \frac{1}{\frac{p_\theta(\mathbf{x}|\mathbf{z})p(\mathbf{z})}{M q_\phi(\mathbf{z}|\mathbf{x})}}} = \frac{1}{1 + \frac{M q_\phi(\mathbf{z}|\mathbf{x})}{p_\theta(\mathbf{x}|\mathbf{z})p(\mathbf{z})}} \in (0, 1) \tag{9}$$

This approximation effectively addresses the first challenge.

For the second challenge, where the proposal distribution $q_\phi(\mathbf{z}|\mathbf{x})$ is implicitly modeled by a neural network $\phi$, we aim to estimate it using adversarial training. Specifically, we address the challenge of computing the term $\log p(\mathbf{z}) - \log q_\phi(\mathbf{z}|\mathbf{x})$ by introducing an additional discriminative network

$T(\mathbf{x}, \mathbf{z})$, which distinguishes between pairs $(\mathbf{x}, \mathbf{z})$ sampled from the true joint distribution $p(\mathbf{x}, \mathbf{z})$, and pairs $(\mathbf{x}, \mathbf{z})$ sampled using the implict proposal distribution $q_\phi(\mathbf{z}|\mathbf{x})$. The additional objective $D(T)$ for this discriminator $T(\mathbf{x}, \mathbf{z})$ is defined as

$$D(T) = \mathbb{E}_{p(\mathbf{x})}\mathbb{E}_{q_\phi(\mathbf{z}|\mathbf{x})}\left[\log \sigma(T(\mathbf{x}, \mathbf{z}))\right] + \mathbb{E}_{p(\mathbf{x})}\mathbb{E}_{p(\mathbf{z})}\left[\log(1 - \sigma(T(\mathbf{x}, \mathbf{z})))\right], \quad (10)$$

where $\sigma(t) = \frac{1}{1+e^{-t}}$ denotes the sigmoid function. By Goodfellow et al. (2014); Mescheder et al. (2017), the optimal discriminator $T^*(\mathbf{x}, \mathbf{z})$ for this objective is given by

$$T^*(\mathbf{x}, \mathbf{z}) = \arg\max_T D(T) = \log q_\phi(\mathbf{z}|\mathbf{x}) - \log p(\mathbf{z}). \quad (11)$$

We see that $T^*$ can be directly substituted into Equation (9) to compute the implicit proposal distribution,

$$a(\mathbf{z}; \mathbf{x}, \theta, \phi) = \frac{1}{1 + \frac{M \exp(T^*(\mathbf{x}, \mathbf{z}))}{p_\theta(\mathbf{x}|\mathbf{z})}}. \quad (12)$$

As a result, we can effectively perform rejection sampling even in the absence of explicit analytical forms for the proposal distribution.

For estimating density ratios of non-analytical distributions, numerous methods are available. In this paper, as the first effort to combine rejection sampling with implicit posterior distributions, we employ adversarial training due to its effective use of Nash equilibrium concepts (Goodfellow et al., 2014; Mescheder et al., 2017). Additionally, because the expectation in Equation (10) is on the outermost layer, Monte Carlo estimation remains unbiased and is suitable for mini-batch algorithms. We plan to explore the integration of other density ratio estimation methods into our approach in future work.

### 3.3 IMPLICIT RESAMPLING EVIDENCE LOWER BOUND (IR-ELBO)

Unlike in traditional implicit variational inference approaches, as our variational distribution we use the distribution resampled via rejection sampling, denoted as $r_{\theta,\phi}$,

$$r_{\theta,\phi}(\mathbf{z}|\mathbf{x}) = \frac{q_\phi(\mathbf{z}|\mathbf{x})a(\mathbf{z}; \mathbf{x}, \theta, \phi)}{Z_{\theta,\phi}(\mathbf{x})}, \quad (13)$$

where $Z_{\theta,\phi}(\mathbf{x}) = \mathbb{E}_{q_\phi(\mathbf{z}|\mathbf{x})}[a(\mathbf{z}; \mathbf{x}, \theta, \phi)]$. To sample from $r_{\theta,\phi}$, we follow the sampling procedure defined in Algorithm 1. First, we use a neural network parameterized by $\eta$ to represent the discriminative network $T_\eta(\mathbf{x}, \mathbf{z})$. By using gradient-based optimization, we obtain its optimal value $T_\eta^*(\mathbf{x}, \mathbf{z})$. Then, through an accept-reject step, we resample from the implicit proposal distribution $r_{\theta,\phi}$.

Then we define the Implicit Resampling Evidence Lower Bound (IR-ELBO) on the marginal log-likelihood of $\mathbf{x}$. This involves using the implicit distribution as the proposal distribution and the resampled distribution $r_{\theta,\phi}$ as the variational distribution. By Jensen's inequality, we have,

$$\log p_\theta(\mathbf{x}) \geq \mathbb{E}_{r_{\theta,\phi}(\mathbf{z}|\mathbf{x})}\left[\log\frac{p_\theta(\mathbf{x}, \mathbf{z})}{r_{\theta,\phi}(\mathbf{z}|\mathbf{x})}\right] = \mathbb{E}_{r_{\theta,\phi}(\mathbf{z}|\mathbf{x})}\left[\log\frac{p_\theta(\mathbf{x}, \mathbf{z})Z_{\theta,\phi}(\mathbf{x})}{q_\phi(\mathbf{z}|\mathbf{x})a(\mathbf{z}; \mathbf{x}, \theta, \phi)}\right]. \quad (14)$$

In Equation (14), we can use the discriminative network $T_\eta^*(\mathbf{x}, \mathbf{z})$ described in Algorithm 1 to compute the probability density function of the implicit distribution. Using Equation (9) and Equation (11), we therefore have that

$$\mathbb{E}_{r_{\theta,\phi}(\mathbf{z}|\mathbf{x})}\left[\log\frac{p_\theta(\mathbf{x}, \mathbf{z})Z_{\theta,\phi}(\mathbf{x})}{q_\phi(\mathbf{z}|\mathbf{x})a(\mathbf{z}; \mathbf{x}, \theta, \phi)}\right] = \mathbb{E}_{r_{\theta,\phi}(\mathbf{z}|\mathbf{x})}\left[\log\left(\frac{p_\theta(\mathbf{x}|\mathbf{z})}{\exp\left(T_\eta^*(\mathbf{x}, \mathbf{z})\right)} + M\right)\right] + \log Z_{\theta,\phi}(\mathbf{x}). \quad (15)$$

For the term $\log Z_{\theta,\phi}(\mathbf{x})$, we can again use Jensen's inequality to define a lower bound,

$$\log Z_{\theta,\phi}(\mathbf{x}) \geq \mathbb{E}_{q_\phi(\mathbf{z}|\mathbf{x})}[\log a(\mathbf{z}; \theta, \phi)] = \mathbb{E}_{q_\phi(\mathbf{z}|\mathbf{x})}\left[\log\frac{p_\theta(\mathbf{x}|\mathbf{z})}{p_\theta(\mathbf{x}|\mathbf{z}) + M\exp\left(T_\eta^*(\mathbf{x}, \mathbf{z})\right)}\right]. \quad (16)$$

Substituting the lower bound for $\log Z_{\theta,\phi}(\mathbf{x})$ from Equation (16) into Equation (15) yields the final loss function, which we call the IR-ELBO. Analogous to Equation (10), since the expectation is applied to the outermost layer, the Monte Carlo approximation of this objective function remains

---

**Algorithm 1:** Sampler for $r_{\theta,\phi}(\mathbf{z}|\mathbf{x})$

---

**Require:** $a_{\theta,\phi}(\mathbf{z};\theta,\phi)$, $q_\phi(\mathbf{z}|\mathbf{x})$
**Ensure:** $\mathbf{z} \sim r_{\theta,\phi}(\mathbf{z}|\mathbf{x})$
 1: Perform gradient ascent on $D(T_\eta)$ in Equation (10) with respect to $\eta$ to obtain $T_\eta^*$
 2: **while** True **do**
 3:    $\mathbf{z} \leftarrow$ sample from implict proposal $q_\phi(\mathbf{z}|\mathbf{x})$ by Equation (4)
 4:    Compute acceptance probability $a(\mathbf{z};\mathbf{x},\theta,\phi)$ by Equation (12)
 5:    Sample uniform $u \sim \mathcal{U}[0,1]$
 6:    **if** $u < a(\mathbf{z};\mathbf{x},\theta,\phi)$ **then**
 7:      Output sample $\mathbf{z}$
 8:    **end if**
 9: **end while**

---

**Algorithm 2:** Implicit Variational Rejection Sampling (IVRS)

---

**Require:** Data $\mathbf{x}$, model parameters $\theta$, neural network parameters $\phi$ and $\eta$
**Ensure:** Optimized parameters $\theta^*$ and $\phi^*$
 1: **Sample Generation:** Generate samples $\{\mathbf{z}_i\}_{i=1}^Q$ from implicit proposal $q_\phi(\mathbf{z}|\mathbf{x})$.
 2: **Density Ratio:** Use discriminator network $T_\eta(\mathbf{x},\mathbf{z})$ to estimate density ratio for each $\mathbf{z}_i$.
 3: **Rejection Sampling:** Accept/reject $\mathbf{z}_i$ based on acceptance function $a(\mathbf{z}_i;\mathbf{x}_i,\theta,\phi)$. (Alg. 1)
 4: **ELBO Optimization:** Compute the IR-ELBO by Equation (16).
 5: Update $\theta \leftarrow \theta + \alpha\nabla_\theta$IR-ELBO.
 6: Update $\phi \leftarrow \phi + \beta\nabla_\phi$IR-ELBO.
 7: Repeat steps 1 to 6 until convergence.

---

unbiased and is appropriate for mini-batch algorithms. Samples from the implicit distribution can be directly obtained, and by adjusting the parameter $M$ they can be resampled.

Using the above derivation, we propose a new inference method for generative models called Implicit Variational Rejection Sampling (IVRS), which combines the strengths of implicit distributions and rejection sampling to achieve a more accurate posterior approximation. The algorithm is shown in Algorithm 2. Optimization of the discriminator network $T_\eta(\mathbf{x},\mathbf{z})$ is reflected in both Algorithm 1 and Algorithm 2; these steps can be merged in practice.

### 3.4 ANALYSIS

In this section, we briefly analyze the properties of the resampling distribution $r_{\theta,\phi}(\mathbf{z}|\mathbf{x})$ and show that it is indeed a better approximation compared to the implicit proposal distribution $q_\phi(\mathbf{z}|\mathbf{x})$. To that end, we directly compute the KL divergence between $r_{\theta,\phi}(\mathbf{z}|\mathbf{x})$ and the true posterior $p_\theta(\mathbf{z}|\mathbf{x})$,

$$\mathrm{KL}\left(r_{\theta,\phi}(\mathbf{z}|\mathbf{x})||p_\theta(\mathbf{z}|\mathbf{x})\right) = \int \frac{q_\phi(\mathbf{z}|\mathbf{x})a(\mathbf{z};\mathbf{x},\theta,\phi)}{Z_{\theta,\phi}(\mathbf{x})} \log \frac{q_\phi(\mathbf{z}|\mathbf{x})a(\mathbf{z};\mathbf{x},\theta,\phi)}{Z_{\theta,\phi}(\mathbf{x})p_\theta(\mathbf{z}|\mathbf{x})} d\mathbf{z}. \quad (17)$$

We directly substitute Equation (9) into the above Equation (17). The right-hand side of Equation (17) can then be rewritten as

$$\int \frac{p_\theta(\mathbf{z}|\mathbf{x})q_\phi(\mathbf{z}|\mathbf{x})}{Z_{\theta,\phi}(\mathbf{x})\left(p_\theta(\mathbf{z}|\mathbf{x}) + Mq_\phi(\mathbf{z}|\mathbf{x})\right)} \log \frac{q_\phi(\mathbf{z}|\mathbf{x})}{Z_{\theta,\phi}(\mathbf{x})\left(p_\theta(\mathbf{z}|\mathbf{x}) + Mq_\phi(\mathbf{z}|\mathbf{x})\right)} d\mathbf{z}. \quad (18)$$

We observe that $\mathrm{KL}\left(r_{\theta,\phi}(\mathbf{z}|\mathbf{x}) \parallel p_\theta(\mathbf{z}|\mathbf{x})\right)$ is a monotonically decreasing function of $M$. When $M$ approaches 0, the acceptance rate approaches 1 and $r_{\theta,\phi}$ approaches $q_\phi(\mathbf{z}|\mathbf{x})$. Conversely, when $M$ approaches $+\infty$, the acceptance rate approaches 0 and $r_{\theta,\phi}$ approaches the true posterior distribution $p_\theta(\mathbf{z}|\mathbf{x})$. It follows that in our algorithm, $r_{\theta,\phi}$ provides a more accurate approximation compared to the implicit proposal distribution $q_\phi(\mathbf{z}|\mathbf{x})$, which in turn helps achieve a tighter variational lower bound. However there is a trade-off: If the rejection rate is too high, this can hinder the model's ability to efficiently obtain samples. Therefore, an appropriate choice of $M$ is critical to ensure that the rejection rate remains a practically suitible value. In our experiments we empirically determine $M$ by cross validation.

## 4    RELATED WORK

The main idea of implicit variational inference is to transform a simple base distribution into a more expressive one using a deep neural network (Mescheder et al., 2017; Huszár, 2017; Titsias & Ruiz, 2019; Shi et al., 2017). To avoid density ratio estimation, semi-implicit variational inference has been proposed, where the variational distributions are formed through a semi-implicit hierarchical construction, and surrogate ELBOs (asymptotically unbiased) are employed for training (Yin & Zhou, 2018; Molchanov et al., 2019; Moens et al., 2021; Lim & Johansen, 2024; Yu & Zhang, 2023; Cheng et al., 2024). These methods have made improvements to implicit variational inference or semi-implicit variational inference at various levels. However, these methods do not address the limited expressiveness of neural networks, which may still fall short in certain scenarios. Our work aims to address this issue by proposing an implicit rejection sampling algorithm.

On the other hand, rejection sampling is a classical method to generate samples from a distribution using samples drawn from a different distribution (Gilks & Wild, 1992; Grover et al., 2018; Azadi et al., 2018; Stimper et al., 2022; Verine et al., 2024). Specifically, Grover et al. (2018) and Stimper et al. (2022) have embedded latent rejection sampling within their training processes, applying it within a variational inference and a normalizing flow framework, respectively. We acknowledge their contributions; however, in this work, we address a different problem—leveraging implicit distributions as proposal distributions for rejection sampling and variational inference, creating a novel variational inference algorithm. All experiments were conducted on a single RTX 4090.

## 5    EXPERIMENTS

In this section, we compare IVRS with other ELBO-based methods, including the original Adversarial Variational Bayes (AVB) (Mescheder et al., 2017) and Semi-Implicit Variational Inference (SIVI) (Yin & Zhou, 2018), across a range of unsupervised inference tasks. We first demonstrate the effectiveness of our method and highlight the role of rejection sampling through several toy examples. Additionally, we compare the performance of IVRS against both baseline methods on various Bayesian inference tasks, including regression with Bayesian Neural Networks (BNNs), as well as a VAE task. We also evaluate other variants of SIVI like UIVI (Titsias & Ruiz, 2019). Due to the parallel computation capabilities of the deep learning with PyTorch (Paszke et al., 2019), which can be utilized for the parallel processing of the acceptance-rejection sampling step in Algorithm 1, our algorithm achieves greater computational efficiency. We use the Adam optimizer (Kingma & Ba, 2015) and empirically select $M$ for training.

### 5.1    DENSITY ESTIMATION OF TOY DATASETS

We first apply rejection sampling with an implicit distribution as the proposal to approximate six synthetic distributions: A 1-dimensional Gaussian distribution, a 1-dimensional Laplace distribution, a 1-dimensional bimodal Gaussian Mixture Model (GMM), a 2-dimensional banana-shaped distribution, a 2-dimensional X-shaped mixture of Gaussians, and a 2-dimensional bimodal GMM. The six density functions are provided in Table 1 along with the performance results described further below.

The dimension of $\epsilon$ was set to 10, and the network approximation $f_\phi$ was parameterized by a 4-layer MLP with layer widths $[20, 40, 20, 2]$. The output of $f_\phi$ was then combined with Gaussian noise. Since this task is straightforward, we adopt a Monte Carlo estimator to estimate $q_\phi$ and $\mathrm{KL}(q_\phi||p_{\mathrm{tar}})$ for gradient-based optimization. The key difference in our method is that the trained neural network was used as the proposal distribution, followed by rejection sampling using the acceptance probability function defined in Equation (9). Following Yin & Zhou (2018), we used 100 iterations for each inner-loop of Monte Carlo sampling to estimate the entropy of the implicit distribution. All methods were trained with 50,000 parameter updates.

Figure 1 shows the contour plots of the synthetic distributions, along with kernel density estimates from samples drawn from the trained implicit distributions $q_\phi$. Additionally, Figure 1 compares the distributions before and after applying rejection sampling. The results confirm our hypothesis that, even for simple 1-dimensional and 2-dimensional distributions, the neural network's approximation ability is limited, as evidenced by the misalignment with the target distribution's contour. However, after applying rejection sampling, our method demonstrates improved approximation with better

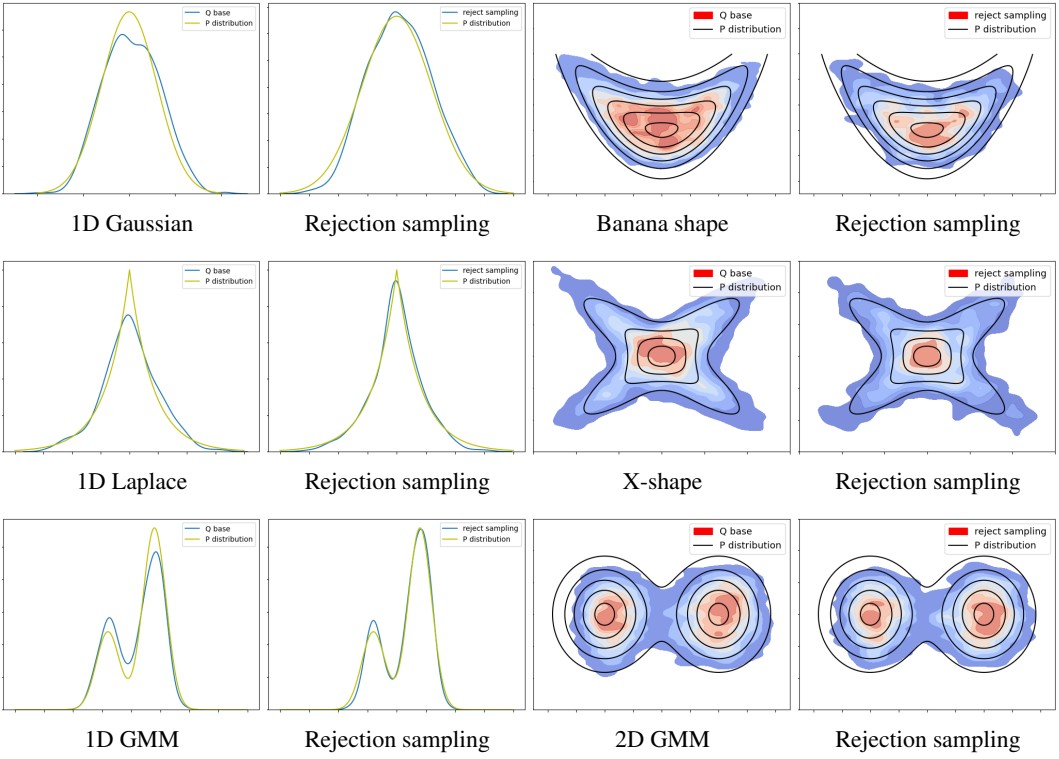

Figure 1: Density estimation tasks for 1D and 2D toy datasets. The subplots sequentially show the kernel density curves estimated using the original implicit distribution $Q$ base for the target distribution $P$, as well as the curves after applying rejection sampling. Quantitative results in Table 1 show the improvement in cross-entropy provided by the rejection sampling step, verifying the visual improvement.

| Distribution | $p_{\mathrm{tar}}(\mathbf{z})$ | $-\mathbb{E}_{q_\phi(\mathbf{z})}[\log p_{\mathrm{tar}}(\mathbf{z})]$ | w/ rejection sampling | $M$ |
|---|---|---|---|---|
| 1D Gaussian | $\mathcal{N}(0,3)$ | $-0.095 \pm 0.001$ | $-0.102 \pm 0.001$ | 0.1 |
| 1D Laplace | $\mathrm{Laplace}(0,2)$ | $-0.107 \pm 0.001$ | $-0.112 \pm 0.001$ | 0.1 |
| 1D GMM | $0.3\mathcal{N}(-2,1) + 0.7\mathcal{N}(2,1)$ | $-0.101 \pm 0.000$ | $-0.104 \pm 0.000$ | 0.1 |
| 2D Banana | $\mathcal{N}(z_1|z_2^2/4,1)\mathcal{N}(z_2|0,4)$ | $1.235 \pm 0.005$ | $1.189 \pm 0.003$ | 500 |
| 2D X-shape | $0.5\mathcal{N}\left(0,\begin{bmatrix} 2 & \pm 1.8 \\ \pm 1.8 & 2 \end{bmatrix}\right)$ | $0.744 \pm 0.008$ | $0.682 \pm 0.005$ | 500 |
| 2D GMM | $0.5\mathcal{N}(-2,I) + 0.5\mathcal{N}(2,I)$ | $1.292 \pm 0.006$ | $1.214 \pm 0.004$ | 500 |

Table 1: Comparison of cross-entropy values before and after rejection sampling for different targets using implicit distributions as proposal distributions. Rejection sampling can provide improved approximate posterior samples.

alignment to the target distribution and more accurate matching of the distribution's peaks. Due to the challenges in normalizing the distribution after rejection sampling, we instead report the cross-entropy between the target distributions and the approximate distributions in Table 1. We conducted 10 runs and report the mean and standard deviation of the cross-entropy values. As shown, our method outperforms across all toy target distributions by reducing the cross-entropy, further validating its effectiveness.

## 5.2 BAYESIAN NEURAL NETWORK

We compare our method with AVB, SIVI, and several SIVI variants, including UIVI, SIVI-SM (Yu & Zhang, 2023), and KSIVI (Cheng et al., 2024), for sampling from the posterior of a Bayesian neural

|  | BOSTON | | CONCRETE | | PROTEIN | |
|---|---|---|---|---|---|---|
|  | NLL ($\downarrow$) | RMSE ($\downarrow$) | NLL ($\downarrow$) | RMSE ($\downarrow$) | NLL ($\downarrow$) | RMSE ($\downarrow$) |
| AVB | $2.489 \pm 0.02$ | $2.685 \pm 0.03$ | $3.406 \pm 0.01$ | $7.091 \pm 0.03$ | $2.969 \pm 0.05$ | $4.670 \pm 0.04$ |
| SIVI | $2.481 \pm 0.00$ | $2.621 \pm 0.02$ | $3.337 \pm 0.00$ | $6.932 \pm 0.02$ | $2.967 \pm 0.03$ | $4.669 \pm 0.02$ |
| UIVI | $2.490 \pm 0.02$ | $2.617 \pm 0.03$ | $3.331 \pm 0.01$ | $6.806 \pm 0.02$ | $2.973 \pm 0.03$ | $4.671 \pm 0.02$ |
| SIVI-SM | $2.542 \pm 0.01$ | $2.785 \pm 0.03$ | $3.229 \pm 0.01$ | $5.973 \pm 0.04$ | $3.047 \pm 0.00$ | $5.087 \pm 0.01$ |
| KSIVI | $2.506 \pm 0.01$ | $2.555 \pm 0.02$ | $3.309 \pm 0.01$ | $5.750 \pm 0.03$ | $3.034 \pm 0.04$ | $5.027 \pm 0.01$ |
| IVRS | $\mathbf{2.365 \pm 0.03}$ | $\mathbf{2.421 \pm 0.03}$ | $\mathbf{2.964 \pm 0.01}$ | $\mathbf{5.68 \pm 0.04}$ | $\mathbf{2.794 \pm 0.04}$ | $\mathbf{4.601 \pm 0.03}$ |

|  | POWER | | WINE | | YACHT | |
|---|---|---|---|---|---|---|
|  | NLL ($\downarrow$) | RMSE ($\downarrow$) | NLL ($\downarrow$) | RMSE ($\downarrow$) | NLL ($\downarrow$) | RMSE ($\downarrow$) |
| AVB | $2.795 \pm 0.02$ | $3.865 \pm 0.02$ | $0.905 \pm 0.01$ | $0.609 \pm 0.00$ | $1.751 \pm 0.06$ | $1.567 \pm 0.04$ |
| SIVI | $2.791 \pm 0.00$ | $3.861 \pm 0.01$ | $0.904 \pm 0.00$ | $0.597 \pm 0.00$ | $1.721 \pm 0.03$ | $1.505 \pm 0.07$ |
| UIVI | $2.794 \pm 0.00$ | $3.863 \pm 0.02$ | $0.907 \pm 0.00$ | $0.613 \pm 0.00$ | $1.808 \pm 0.03$ | $1.569 \pm 0.05$ |
| SIVI-SM | $2.822 \pm 0.00$ | $4.009 \pm 0.00$ | $0.916 \pm 0.00$ | $0.615 \pm 0.00$ | $1.432 \pm 0.01$ | $\mathbf{0.884 \pm 0.01}$ |
| KSIVI | $2.797 \pm 0.00$ | $3.868 \pm 0.01$ | $0.901 \pm 0.00$ | $0.595 \pm 0.00$ | $1.752 \pm 0.03$ | $1.237 \pm 0.05$ |
| IVRS | $\mathbf{2.670 \pm 0.01}$ | $\mathbf{3.684 \pm 0.03}$ | $\mathbf{0.900 \pm 0.00}$ | $\mathbf{0.591 \pm 0.00}$ | $\mathbf{1.421 \pm 0.03}$ | $1.065 \pm 0.04$ |

Table 2: Quantitative results for six UCI regression tasks. More accurate posterior sampling allows IVRS to outperform several other VI approximations for learning the same Bayesian neural network.

network. In this scenario, the latent model variables $\mathbf{z}$ correspond to the BNN weights. We use several UCI datasets to perform these experiments. We utilize a two-layer network with 50 hidden units and ReLU activation functions. Each dataset is randomly partitioned, with 90% used for training and 10% for testing. Both the proposal distribution $\phi$ and the discriminator $\eta$ are modeled using four-layer fully connected neural networks. The results are averaged over 10 random trials.

Table 2 presents the average test root mean squared error (RMSE) and negative log-likelihood (NLL) along with their standard deviations. The six UCI datasets considered are indicated by their common names. These results demonstrate that IVRS achieves comparable or superior performance to all baselines, indicating that rejection sampling provides a more accurate representation of the BNN model variables.

## 5.3 VARIATIONAL AUTOENCODER TASK ON THE MNIST DATASET

Variational Autoencoders (VAEs) (Kingma & Welling, 2013), a popular method for unsupervised feature learning and dimensionality reduction, aim to infer the encoder parameter $\phi$ and decoder parameter $\theta$ by maximizing the Evidence Lower Bound (ELBO) as defined in Equation (2). Traditional VAEs utilize Gaussian distributions and amortized inference to approximate complex posterior distributions $q_\phi$. To enhance this process, various approaches such as Adversarial Variational Bayes (AVB) and Semi-Implicit Variational Inference (SIVI) have been proposed, leveraging adversarial training and semi-implicit hierarchical structures, respectively.

To evaluate the effectiveness of our proposed Implicit Variational Rejection Sampling (IVRS) model, we conducted experiments on the MNIST dataset. We trained our model on 60,000 training samples and evaluated its performance using 10,000 test samples. The encoder is designed as a two-layer convolutional neural network (CNN), while the decoder consists of a four-layer transposed convolution network. The encoder maps the original images to the latent space, and the decoder reconstructs the images. Figure 2 presents sampling examples of 16 images from the test set, demonstrating that our method generates images with a closer resemblance to the ground truth compared to the baseline AVB method, owing to the improved sampling quality introduced by rejection sampling.

To further assess the efficiency of our model, we conducted an empirical time analysis comparing IVRS to the baseline AVB on the MNIST dataset. With a batch size of 64 during training, we observed that our model incurs only a slight increase in computational cost due to the GPU-accelerated parallel sampling, as shown in Table 4. This highlights the efficiency of our approach.

We also benchmarked our model against several well-established methods in this task, including Importance Weighted Autoencoders (IWAE) (Burda et al., 2015), Hamiltonian Variational Inference (HVI) (Salimans et al., 2015), and Normalizing Flows (NF) (Rezende & Mohamed, 2015). With recent advancements in deep learning, many approaches have adopted deep architectures for the

| Methods | $-\log p(\mathbf{x})$ |
|---|---|
| **Results from Burda et al. (2015)** | |
| VAE + IWAE | 86.76 |
| IWAE + IWAE | 84.78 |
| **Results from Salimans et al. (2015)** | |
| VAE + HVI (1 leapfrog step) | 88.08 |
| VAE + HVI (4 leapfrog steps) | 86.40 |
| VAE + HVI (8 leapfrog steps) | 85.51 |
| **Results from Rezende & Mohamed (2015)** | |
| VAE + NICE Dinh et al. (2014) (k = 80) | $\leq 87.2$ |
| VAE + NF (k = 40) | $\leq 85.7$ |
| VAE + NF (k = 80) | $\leq 85.1$ |
| **Results from Gregor et al. (2015)** | |
| NADE | 88.33 |
| DBM 2hl | $\approx 84.62$ |
| DBN 2hl | $\approx 84.55$ |
| EoNADE-5 2hl (128 orderings) | 84.68 |
| DARN 1hl | $\approx 84.13$ |
| **Results from Sønderby et al. (2016)** | |
| Auxiliary VAE ($L = 1$, IW = 1) | $\leq 84.59$ |
| **Results from Mescheder et al. (2017)** | |
| VAE + IAF Kingma et al. (2016) | $\approx 84.9 \pm 0.3$ |
| AVB | $\approx 83.7 \pm 0.3$ |
| **Results from Yin & Zhou (2018)** | |
| SIVI (3 stochastic layers) + IW($\tilde{K} = 10$) | 83.25 |
| **Results from Hazami et al. (2022)** | |
| PixelVAE++ Sadeghi et al. (2019) | 78.00 |
| Locally Masked PixelCNN Jain et al. (2020) | 77.58 |
| NVAE Vahdat & Kautz (2020) | 78.01 |
| CR-NVAE Sinha & Dieng (2021) | **76.93**$^{\star}$ |
| Efficient-VDVAE Hazami et al. (2022) | 79.09 |
| **IVRS** | 81.78 |
| **IVRS**+NVAE Vahdat & Kautz (2020) | **77.36** |

Table 3: Comparison of reported Negative Log Evidence (NLE) values across different algorithms on the MNIST dataset. An asterisk ($\star$) indicates results obtained with data augmentation.

VAE encoder to achieve more effective feature extraction. Therefore, we also report comparisons with recent deep architecture-based VAE improvements such as NVAE (Vahdat & Kautz, 2020), CR-NVAE (Sinha & Dieng, 2021), and Efficient-VDVAE (Hazami et al., 2022).

As shown in Table 3, our IVRS method achieves a Negative Log Evidence (NLE) score of 81.78, surpassing traditional variational inference methods like SIVI and AVB. Additionally, we integrated IVRS with the NVAE architecture, further boosting performance, achieving an NLE of 77.36, which is an improvement over the original NVAE's 78.01. Although CR-NVAE (Sinha & Dieng, 2021) demonstrated slightly better results, it relies on additional data augmentation techniques. While data augmentation is highly effective in preventing overfitting on datasets like CIFAR-10, our focus is on demonstrating the competitiveness of rejection sampling in enhancing implicit variational inference.

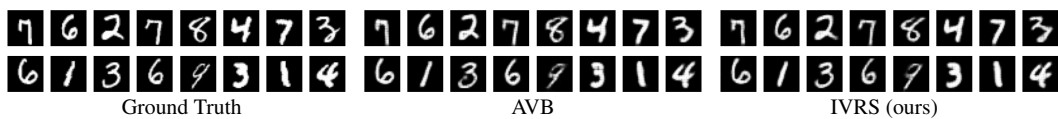

Ground Truth     AVB     IVRS (ours)

Figure 2: Examples of MNIST images generated using the VAE learned by AVB and IVRS.

| Method | Batch Size | Iterations | Time/epoch | Time/iter |
|---|---|---|---|---|
| AVB | 64 | 938 | 19.11s | 0.020s |
| IVRS (Ours) | 64 | 938 | 24.72s | 0.026s |

Table 4: Empirical time analysis comparing the AVB method and our proposed IVRS on the MNIST dataset.

| Methods | bpd |
|---|---|
| VAE + IAF Sønderby et al. (2016) | 3.11 |
| BIVA Maaløe et al. (2019) | 3.08 |
| DVAE Vahdat et al. (2018) | 3.38 |
| $\delta$-VAE Razavi et al. (2019) | 2.83 |
| PixelVAE++ Sadeghi et al. (2019) | 2.90 |
| Locally Masked PixelCNN Jain et al. (2020) | 2.89 |
| MAE Ma et al. (2019) | 2.95 |
| NVAE Vahdat & Kautz (2020) | 2.91 |
| VDVAE Child (2020) | 2.87 |
| Efficient-VDVAE Hazami et al. (2022) | 2.87 |
| CR-NVAE Sinha & Dieng (2021) | **2.51**$^\star$ |
| **IVRS**+NVAE | **2.76** |

Table 5: Comparison of reported Bits per Dimension (bpd) values among various algorithms for the CIFAR-10 dataset.

### 5.4 ADDITIONAL RESULTS ON CIFAR-10 DATASET

To further validate the effectiveness of the IVRS method on high-dimensional image datasets, we conducted experiments on the CIFAR-10 dataset using the VAE task and compared our model against baseline methods. The CIFAR-10 dataset comprises 50,000 training images and 10,000 test images, with each image being $32 \times 32$ pixels and containing 3 color channels (RGB) across 10 classes. Given the increased complexity and higher dimensionality compared to MNIST, we adapted the VAE architecture by directly employing the encoder from the deep architecture NVAE (Vahdat & Kautz, 2020).

For evaluation, we used bits per dimension (bpd), a standard metric for high-dimensional image datasets. Table 5 presents the bpd scores for various variational inference methods on CIFAR-10. As shown, our proposed IVRS method achieves a lower bpd score compared to most classic VAE approaches, indicating improved density estimation quality. Specifically, our IVRS method reduced the bpd to 2.76, outperforming most methods except for CR-NVAE (Sinha & Dieng, 2021), which leverages additional data augmentation. This result demonstrates the advantage of incorporating rejection sampling to obtain a tighter variational lower bound. Moreover, despite the additional step of rejection sampling, the increase in computational time remains minimal, underscoring the practical efficiency of our approach.

## 6 LIMITATIONS

Despite the clear advantages of IVRS, which combines the accuracy of rejection sampling with the flexibility of implicit distributions, there are some limitations. One key limitation is the need for empirical hand-tuning of the hyperparameter $M$ in the acceptance probability function. Additionally, while our experiments have shown improved bpd scores on standard datasets like CIFAR-10, further testing on more complex and higher-dimensional datasets is necessary to fully assess the robustness of our approach. We leave these aspects for future work.

## 7 CONCLUSION

We have introduced Implicit Variational Rejection Sampling (IVRS), a novel posterior approximation method that integrates the flexibility of implicit distributions with the rigor of rejection sampling to enhance variational inference. By utilizing the Implicit Resampled Evidence Lower Bound (IR-ELBO), IVRS achieves a more accurate and efficient posterior approximation. Our experimental results demonstrate the effectiveness of IVRS across various tasks, including regression problems and VAE-based image modeling, achieving superior performance and validating the robustness of our approach.

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
