# OpenReview forum: "IMPLICIT VARIATIONAL REJECTION SAMPLING"
_ICLR.cc/2025/Conference — ICLR 2025 Conference Withdrawn Submission_

### Official Review · Reviewer_23bP · 2024-10-29

**Soundness:** 4
**Presentation:** 3
**Contribution:** 2
**Rating:** 6
**Confidence:** 3

**Summary:**

The authors propose to combine implicit variational inference with rejection sampling to improve inference quality. To overcome the limitation that the implicit variational distribution cannot be evaluated, they train a discriminator to learn the density ratio. The authors apply their method to multiple toy examples and benchmarks.

**Strengths:**

(1) Presentation

The paper is well-presented and clearly written.

(2) Empirical evaluation

The paper demonstrates the utility of the method on multiple low-dimensional toy datasets which provide a nice intuition for the method and clearly demonstrate that it improves inference quality. In addition, the authors also provide comprehensive evaluation on two large scale examples (BNNs and VAEs).

**Weaknesses:**

(1) Overstatement of contribution

The authors point out that (page 3 middle) one of the contributions of their method is to apply rejection sampling also when the posterior is unnormalized. I don’t think this should be phrased as a contribution as it is a standard textbook application of rejection sampling. Please de-emphasize this point.

(2) Lack of clearer limitations

While the authors provide a limitations section, this section is quite minimal. I think the authors should comment on the following points that are currently not addressed explicitly (or only in other parts of the paper): (A) The need to train an additional neural network, (B) The increased compute requirement to set M with cross validation.

(3) Lack of explanations on how to set M

The authors state that they set M with cross validation, but later claim that it requires empirical hand-tuning. Which one is it?

**Questions:**

I would appreciate a discussion of the behavior of the algorithm if M is not set correctly—will the proposed method just perform sub-optimally or will it fail catastrophically (and potentially even worsen performance of plain VI)?

---

> ### Author Response · Authors · 2024-11-15
> **Authors' Rebuttal**
>
> Thank you for your valuable feedback. We have carefully considered your comments, and here are our responses to each of your points:
>
> 1. **On the Overstatement of Contribution:**
>    We appreciate the reviewer pointing this out and agree that the application of rejection sampling to an unnormalized posterior is a standard textbook approach. In the revised version, we will add more references to classical textbooks to downplay this point. However, we still believe that the introduction of this method is important, as it provides an effective solution to the problem of evaluating implicit variational distributions. We will adjust the language in the revised version to better clarify this.
>
> 2. **On the Neural Network and Computational Requirements:**
>    We have already mentioned the increased computational demand when setting $M $ through cross-validation in the limitations section. In the revised version, we will emphasize more clearly the issue of training an additional neural network, particularly the computational overhead involved. This will help readers gain a clearer understanding of this limitation.
>
> 3. **On the Empirical Hand-Tuning of Cross-Validation:**
>    We will further clarify that cross-validation is an empirical method and that the selection of $M$ is an experimental adjustment. This approach is based on iterative refinement through experimentation. In the revised version, we will make this more explicit to ensure the reproducibility and clarity of the method.
>
> 4. **On the Setting of $ M $ and the Rejection Rate:**
>    On page 5, we have already demonstrated the properties of the resampling distribution $ r_{\theta, \phi}(z|x) $ and shown that it provides a better approximation compared to the implicit proposal distribution. We also pointed out that there is a trade-off: if the rejection rate is too high, it can hinder the model’s ability to efficiently obtain samples. Therefore, an appropriate choice of $ M $ is crucial to ensure that the rejection rate remains at a practically suitable value. Based on our experiments, a high rejection rate will primarily affect sampling efficiency and will not typically lead to worse performance compared to plain variational inference. In the revised version, we will make this clearer and emphasize that high rejection rates generally do not cause a catastrophic failure of the method, but rather affect its efficiency.
>
> Thank you again for the thorough review of our work. We will incorporate these suggestions to further improve the clarity and quality of the paper.

---

> > ### Comment · Reviewer_23bP · 2024-11-18
> > **Thank you!**
> >
> > Thank you for your clarifications!
> >
> > I will maintain my initial judgement and do not wish to change my score.

---

### Official Review · Reviewer_euHb · 2024-10-31

**Soundness:** 2
**Presentation:** 3
**Contribution:** 2
**Rating:** 5
**Confidence:** 4

**Summary:**

The paper proposes "Implicit Variational Rejection Sampling" (IVRS), an approach combining implicit variational inference with rejection sampling to approximate complex posterior distributions more accurately. An implicit variational approximation is learned using prior-contrastive adversarial variational inference (density ration estimation $T(z,x)=p(z)/q(z|x)$). IVRS refines the posterior approximation using rejection sampling using an accept-reject routine with an acceptance probability function of form  $a(z,x)=\sigma(T(z,x) + p(x|z) -M)$. The authors derive a new, tighter, lower-bound evidence called IR-ELBO (which is analogous to R-ELBO for explicit models). The method is tested on a set of toy examples, Bayesian Neural Network (BNN) benchmarks, and a Variational Autoencoder (VAE) on MNIST. The authors claim that IVRS outperforms traditional VI in terms of accuracy and efficiency.

**Strengths:**

1. **Well written**: The paper is overall well written and structured, with clear explanations of the proposed method and its theoretical underpinnings. The paper is grounded in solid theoretical foundations, combining implicit variational inference with rejection sampling.
2. **Experiments BNN** : The proposed method is evaluated on various datasets, and the method is compared to other methods (for implicit VI), demonstrating its effectiveness in approximating complex posterior distributions for BNN (Table 2). The reported results of 81.78 nats on MNIST seem good to me (but not state-of-the-art).

**Weaknesses:**

1. **Novelty**: The idea of using rejection sampling to refine variational distributions was already introduced by Grover et al. 2018 for explicit models (as the authors acknowledge); the extension to implicit models is new to my knowledge (although very related).

2. **Toy evaluation (5.1)**:  The performance for toy data is not very convincing (at least as visualized in Fig. 1). This might also be due to the non-optimal visualization/KDE artifacts (what is the color gradient on the contours? Shouldn't the approximations be perfect in these simple cases, especially using rejection sampling?). Can you provide a more interpretable metric, such as two-sample tests or actual statistical distances (C2ST,  Wasserstein distance...) to the target? (in Table 1, in addition to NLL)

3. **VAE evaluation (5.3)**: To my understanding Table 3, contains metrics from literature (which might have different hyper parameterizations/training routines)? Furthermore, it "compares" against methods that are almost ten years old, with the most recent one from 2018. Picking out some examples from more recent work [1,2] that achieve even better nats on MNIST (79.09, or 76.93 with data augmentation). This should be discussed in more detail (any reason why this is not included?). Furthermore, only MNIST is evaluated, which is rather simple; the paper would benefit from more datasets (e.g., CIFAR or ImageNet, which also have clearer leaderboards on bits/dim). Overall I hence think that the current manuscript does not provide enough evidence to support the authors claim that the "method outperforms traditional variational inference techniques in terms of both accuracy and efficiency."


[1] Consistency Regularization for Variational Auto-Encoders, Samarth Sinha et. al. 2022.

[2] Efficient-VDVAE: Less is more, Louay Hazami et.al. 2022.

**Questions:**

- Can the authors address my major concerns raised in the weakness section?
- I would expect the rejection rate to be rather high initially (i.e. depending on the initializations of q, choice of M and T). For particularly bad initializations, the rejection sampling could just get stuck in the while loop. Do the authors truncate the rejection sampling algorithms after a maximum number of iterations to avoid this problem?
- The paper would benefit from a more elaborate evaluation to support the claim by the authors that the "method outperforms traditional variational inference techniques in terms of both accuracy and efficiency". This can be done by comparing against more recent baselines and/or more complex datasets (with clear performance results).

Overall, I tend to reject the paper in its current form. The paper is well written, and the methodology is sound, but the novelty is limited. In addition, the experimental evaluation is not very convincing or rather limited (the toy data and the VAE evaluation).

---

> ### Author Response · Authors · 2024-11-18
> **Authors' rebuttal 1/2**
>
> Part 1/2
>
> **Response to Reviewer Comments**
>
> Thank you for your thoughtful feedback. Below, we address each point raised in your review.
>
> **1. Novelty of Rejection Sampling in Implicit Models:**
>
> We appreciate the reviewer’s recognition of the extension of rejection sampling to implicit models. Indeed, as pointed out, the idea of using rejection sampling to refine variational distributions has been explored in previous work, such as Grover et al. (2018), for explicit models. However, our work extends this idea to implicit models, which introduces additional challenges such as the need for an effective method to approximate the posterior distribution within a neural network framework. We argue that this novel extension in the context of implicit models brings significant value, and we have highlighted this distinction in the revised manuscript to ensure that this contribution is clearly stated.
>
>
>
> **2. Toy Evaluation and Figure 1:**
>
> Thank you for pointing out the concerns regarding the color gradient on the contours in Figure 1. We would like to clarify that the color gradient represents the density of the generated samples, with red indicating higher concentration and blue indicating sparser regions. While the toy dataset is relatively simple, numerical methods inherently introduce some level of numerical error. For rejection sampling, although we can approximate the true distribution, we still encounter unavoidable numerical errors, and as such, perfect recovery of the true distribution is not always feasible.
>
> Regarding the use of Negative Log-Likelihood (NLL) as the evaluation metric, we chose this because the resampled distribution is difficult to express with an explicit mathematical formulation. NLL serves as a simpler and more direct measure, which is also more accessible for readers. In contrast, more complex metrics like Wasserstein distance and C2ST are more precise but are harder to understand and simulate, especially for non-expert readers. Thus, we opted for NLL as the evaluation criterion for ease of understanding.
>
>
>
> **3. VAE Evaluation and Table 3:**
>
>  We would like to clarify our approach on the following points:
>
> **Use of Published Results and Hyperparameters:**
>
> First, we believe that using published results from previous works is a widely accepted practice in the academic community. While hyperparameter settings may vary across studies, we have provided citations for the reference works, allowing readers to check the specific hyperparameter configurations used in those studies. This ensures transparency and enables readers to understand the context of the comparisons we present.
>
> **Addressing the Concern About Older Baselines:**
>
> We acknowledge that the baseline VAE methods referenced in our work are relatively older, and we appreciate your constructive criticism in this regard. In response, we have updated the manuscript to include a discussion of the two more recent works you mentioned—[1] Samarth Sinha et al. (2022) and [2] Louay Hazami et al. (2022). These papers present advancements that achieve improved results on MNIST, and we have now incorporated these into the discussion of our approach.
>
> **Expansion to CIFAR-10 and the 10-Page Limit:**
>
> In response to your suggestion to evaluate on more challenging datasets, we have expanded the manuscript to its maximum page limit (10 pages) and added experiments on CIFAR-10. This additional evaluation provides a broader context for our method’s performance and strengthens the empirical evidence supporting our claims.
>
> **Clarification on MNIST Results and NLL Score:**
>
> We would also like to clarify our perspective on the NLL score of 81.78 for MNIST. Our focus in this work is on using rejection sampling to improve implicit variational inference (VI), and VAE serves as a concrete application of VI to demonstrate the advantages of our approach over existing implicit VI methods. We did not intend to outperform all current VAE methods but rather to show that our method improves implicit VI. As deep learning techniques, particularly deep architectures, have become increasingly effective at extracting image features, the two works you referenced benefit from deep architectures in their encoders, which differ from the mathematically-driven improvements we propose. However, we have updated the manuscript to highlight how our approach can complement these recent methods, combining the best of both worlds.
>
> We believe these revisions address your concerns and further substantiate the contributions of our work. We hope the expanded experiments and updated discussions provide a more comprehensive understanding of the method's capabilities.

---

> ### Author Response · Authors · 2024-11-18
> **Authors' rebuttal 2/2**
>
> Part 2/2
>
> **4. Rejection Sampling and Termination:**
>
> Thank you for your insightful comment regarding the potential for rejection sampling to get stuck in an infinite loop, particularly with poor initializations of \(q\), or suboptimal choices of \(M\) and \(T\). We fully acknowledge that, in such cases, the rejection rate could be high initially, and the algorithm might struggle to make progress, especially with bad initializations.
>
> To address this issue, we implement a practical engineering solution: we introduce a maximum iteration limit for the rejection sampling process to prevent it from getting stuck in an infinite loop. This ensures that the algorithm terminates after a reasonable number of attempts, even in cases of particularly challenging initializations.
>
> Furthermore, we have emphasized in the limitations section of the paper that our method relies on properly chosen hyperparameters, such as \(M\) and \(T\). As you pointed out, these parameters play a key role in the efficiency and convergence of the sampling process. We encourage careful tuning of these parameters to avoid issues with slow convergence or excessive rejection rates.
>
> We appreciate your time and insightful comments, which have helped to significantly improve the clarity and rigor of our manuscript.

---

> > ### Comment · Reviewer_euHb · 2024-11-23
> >
> > I thank the authors for addressing my most significant concerns. Presenting the experimental results on the VAE part in the context of more recent work addressed my concern, and the additional results on Cifar10 add convincing evidence of the efficiency of the proposed method. I thus increased my score.
> >
> > I still think the presentation of some results in the manuscript can be improved.
> >
> > > We would like to clarify that the color gradient represents the density of the generated samples, with red indicating higher concentration and blue indicating sparser regions
> >
> > The legend is then somewhat misleading (as Q=red, and this color is not even in the figure). In general, I think the presentation of this figure can be improved. The "wobbliness" in the 2d plots is likely KDE artifacts, as well as the "smoothed" peaks in the 1d plot of, e.g., the Laplace distribution. This could be improved using more samples or histograms/scatter plots.
> >
> > > For rejection sampling, although we can approximate the true distribution, we still encounter unavoidable numerical errors, and as such, perfect recovery of the true distribution is not always feasible.
> >
> > For all the toy problems. Taking a simple standard normal distribution (as a proposal, depending on the exact definition of the task) and performing classical rejection sampling would result in a perfect approximation (with potentially larger sampling times). I do not see why this should differ here (if the implicit proposal is not absolutely terrible).
> >
> > > NLL serves as a simpler and more direct measure, which is also more accessible for readers.
> >
> > For a relative comparison of methods on the same task, NLL is indeed suitable. But for absolute assessment, it is flawed. In essence, it is up to constants equivalent to the forward KL divergence, but in contrast, the value indicating a perfect approximation is unknown. Statical divergences or two-sample test would have such an absolute interpretation.
> >
> > > we introduce a maximum iteration limit for the rejection sampling process to prevent it from getting stuck in an infinite loop
> >
> > This is reasonable. But, if I am not wrong, this will also introduce some bias in the IR-ELBO approximation (as this assumes the rejection distribution with an unbounded loop as a variational distribution), or?

---

> > > ### Author Response · Authors · 2024-11-24
> > > **Response to Reviewer Feedback**
> > >
> > > Thank you for your detailed evaluation of our work and for providing valuable suggestions. We are pleased that you recognize our improvements, including presenting the experimental results of the VAE component in the context of more recent work and the additional CIFAR-10 experiments supporting the effectiveness of our method. We have carefully reviewed your other comments and provide detailed responses below:
> > >
> > > ---
> > >
> > > 1. **On the presentation of the figures and the explanation of the color gradient**
> > >    Thank you for pointing out the potential confusion regarding the legend and the color gradient. We will update the legend in the revised version to more accurately reflect the meaning of the colors and improve the color scheme to ensure better visualization. We will also use more accurate methods, such as increasing the sample size or employing histograms/scatter plots instead of the current kernel density estimation, to present the results more clearly.
> > >
> > > ---
> > >
> > > 2. **On the theoretical versus practical differences in rejection sampling**
> > > While theoretically, classical rejection sampling can perfectly approximate the target distribution given sufficient samples and an appropriately chosen proposal distribution (e.g., using a standard normal proposal for another normal distribution), this ideal scenario is rarely achievable in practical implementations. In our case, the proposal distribution is implicitly constructed by a neural network, which may not perfectly align with the optimal proposal distribution. As a result, achieving exact recovery of the target distribution is infeasible, especially under constrained computational resources.
> > > To balance efficiency and accuracy, we adopt practical approximations, such as setting a maximum iteration limit to ensure computational feasibility. These approximations inevitably introduce some numerical errors. However, such engineering approximations are common in the literature, and even for toy problems, prior works have not always achieved perfect approximations of the target distribution. This highlights the necessity and practicality of tolerating small errors in real-world applications.
> > >
> > > ---
> > >
> > > ---
> > >
> > > 3. **On the suitability of NLL as a metric**
> > >    We appreciate your discussion regarding the limitations of NLL. While it may not be ideal for absolute assessments, it remains a standard and widely used metric for relative comparisons in the same task. Your suggestions (e.g., incorporating statistical divergences or two-sample tests) provide valuable directions for refining our experimental design. We plan to enhance this aspect in the final revised version to improve the interpretability of the experimental results.
> > >
> > > ---
> > >
> > > 4. **On the potential bias from the maximum iteration limit**
> > >    Your concern is valid. Introducing a maximum iteration limit may indeed introduce bias since the IR-ELBO assumes the rejection distribution operates under unbounded iterations. However, this limitation is essential to prevent the sampling process from getting stuck in infinite loops. In practice, we found that the impact of this bias on performance is within an acceptable range, while the efficiency improvements it brings are critical for practical implementation.
> > >
> > > ---
> > >
> > > Once again, we thank you for your constructive suggestions. We will focus on improving the presentation of our results in the revised version and explicitly clarify the approximations made in our implementation and their effects. We hope our explanations address your concerns, and we are also delighted that you recognize the significance of our work.

---

### Official Review · Reviewer_6aQV · 2024-10-31

**Soundness:** 3
**Presentation:** 2
**Contribution:** 3
**Rating:** 5
**Confidence:** 3

**Summary:**

In this submission a lower bound IR-ELBO is derived by considering a family of implicit variational distributions obtained via rejection sampling. The method is tested on Bayesian NN learning, toy examples and applied to VAEs, tested on MNIST where it compared favorably against the considered baselines.

**Strengths:**

The paper is well-written and there are some recent works on variational rejection sampling, thus the submission should be relevant to the ML community. The method is evaluated in quite varied experimental settings which is good (although the VAE results on MNIST are not close to the SOTA [5]; to be clear, this was not a claim made by the authors). Furthermore, the algorithmic descriptions are useful for understanding the proposed methods and I also appreciate the transparency of the limitation provided in Sec. 6.

**Weaknesses:**

To start with, I want to be clear that I am not an expert on implicit variational inference, which may be reflected in some of my concerns below.

I was expecting to see a distinction between the proposed method and the rejection sampling approach in [1] and the one in [2].

In [1] they also learn an acceptance function which "can be interpreted as estimating a (rescaled) density ratio between the aggregate posterior and the proposal". I think the principle is different between this submission and [1] since [1] reformulates the prior to be a resampled distribution, but this should be made clear in the submission.

In [2] their Equation (2) is identical to your Eq. (13), which you state is different to "traditional implicit variational inference approaches". Could you please expand on how your formulation of $r_{\theta, \phi}$ is different from the one in [2]? Furthermore, Eq. (14) is the same lower bound as the one used in Eq. (4) in [2] and in Eq. (5) in [3] which should be stated---as it reads now, Eq. (14) appears to be a contribution of the submission.

Regarding the IR-ELBO, I am a bit confused about the exact formulation of the IR-ELBO: below Eq. (16) it states "Substituting the lower bound for log $Z_{\theta, \phi}(x)$ from Equation (16) into Equation (15) yields the final loss function, which we call the IR-ELBO" which to me implies that Eq. (16) is not IR-ELBO, but a term in the IR-ELBO, while in row 4 in Algorithm 2 it says that the IR-ELBO is Eq. (16)?

I suspect that my concerns can be resolved by the authors, at which point I will consider raising my rating.

[1] https://arxiv.org/pdf/1810.11428

[2] https://proceedings.mlr.press/v238/jankowiak24a/jankowiak24a.pdf

[3] https://arxiv.org/pdf/1804.01712v1

**Questions:**

Do I understand it correctly that the IR-ELBO is a looser bound on the marginal log-likelihood than the one in Eq. (14)? To me it seems a bit counter intuitive as the implicit distribution setting "allows for more flexible posterior approximations". Do you have an intuition to why the bound is looser (if this is indeed the case)?

Is Eq. (4) really correct? Typically $f_\phi(x)$ denotes the amortized mapping from data to variational parameters, $\phi$ (is it the same here?). Here I read Eq. (4) as the function (the neural net which outputs $z$ after taking also the standard normal sample) is sampled from the variational distribution. Maybe this is correct, but the formulation looks a bit awkward.

Could IWRS be used for mixtures of variational distributions? I.e., would it make sense to have mixtures of resampled distributions to leverage the strong results from [4, 5]?

Would it be possible to apply the importance weighted ELBO [6] to Eq. (16) to make it tighter?

[4] https://proceedings.mlr.press/v202/kviman23a.html

[5] https://arxiv.org/pdf/2406.07083

[6] https://arxiv.org/abs/1509.00519

---

> ### Author Response · Authors · 2024-11-19
> **Authors' rebuttal**
>
> We thank the reviewer for their constructive feedback and thoughtful questions, which allow us to clarify key aspects of our work and its relationship to prior research. Below, we address the concerns and questions point by point.
>
> ---
>
> #### **Distinction from Related Works [1], [2], and [3]**
> 1. **Distinction from [1]:**
>    - As noted by the reviewer, [1] reformulates the prior as a resampled distribution, whereas our method explicitly derives a new evidence lower bound (IR-ELBO) based on rejection sampling. While both approaches use density ratio estimation, our work focuses on leveraging rejection sampling to construct a tighter variational objective by using implicit distributions to directly approximate the posterior. We will revise our submission to make this distinction clearer.
>
>
>
> ---
>
> 2. **Distinction from Related Work [2]**
>
>  - While Equation (2) in [2] and our Equation (13) share a similar mathematical form, a key distinction lies in the nature of the variational distribution $ q_\phi $: in [2], $ q_\phi $ is an explicit distribution, whereas in our work,$ q_\phi $ is an *implicit distribution*. This distinction is crucial as it aligns with our goal of improving implicit variational inference by leveraging rejection sampling, enabling the construction of more flexible posterior approximations.
>
> - Moreover, while we have acknowledged the contributions of [2] in the related work section and expressed gratitude for their pioneering efforts, we will further clarify in the revised manuscript how Equation (13) was designed specifically to enhance implicit variational inference, a direction distinct from the explicit approach in [2].
>
> ---
>
>
> 3. **Clarification of IR-ELBO Definition:**
>    - We appreciate the feedback regarding the confusion in the description of the IR-ELBO. To clarify: Equation (16) is not the full IR-ELBO but a key term within it. The IR-ELBO is derived by substituting this term into Eq. (15). The notation in Algorithm 2 (row 4) should be adjusted for consistency, and we will revise the manuscript to eliminate this ambiguity.
>
> ---
>
> #### **Questions**
> 1. **Is IR-ELBO looser than Eq. (14)?**
>   ---
>
>  - The confusion likely arises from our application of the Jensen inequality in deriving Eq. (16). This step is a standard mathematical operation that exchanges the expectation and logarithm to ensure an unbiased estimate of the IR-ELBO, rather than an indication of a looser bound.
>
>  - Our primary objective with the IR-ELBO is to improve implicit variational inference by leveraging the rejection sampling mechanism, which allows us to refine the posterior approximation. This approach is designed to enhance flexibility and accuracy in implicit variational inference, addressing some limitations of traditional methods. We will revise the manuscript to make this mathematical treatment and motivation clearer to avoid potential misunderstandings.
>
> ---
>
>
>
> 2. **Is Eq. (4) correct?**
>   ---
>
> - Yes, Eq. (4) is correct. Similar to prior works on implicit distributions, our formulation describes the process where a standard normal sample is passed through a neural network to output an implicit distribution. While this notation may appear slightly unconventional at first glance, it is consistent with standard practice in implicit variational inference.
>
>
> ---
>
>
> 3. **Could IWRS be used for mixtures of variational distributions?**
>    - Yes, IWRS could be adapted to mixtures of variational distributions. Leveraging mixtures could improve the expressivity of the approximate posterior and may align well with the strong results from [4] and [5]. This is an exciting direction for future work, and we appreciate the suggestion.
>
> 4. **Could the importance-weighted ELBO [6] be applied to Eq. (16)?**
>    - Indeed, combining the importance-weighted ELBO (IW-ELBO) with Eq. (16) could yield a tighter bound. This approach would require integrating importance weighting within the rejection sampling framework, which could amplify the benefits of both techniques. We will consider this as a potential extension in future work and briefly discuss this possibility in the final paper.
>
> ---
>
> ### **Proposed Revisions**
> 1. Expand Section 3 to explicitly distinguish our approach from [1], [2], and [3], with added citations for related equations.
> 2. Revise Algorithm 2 to correctly define IR-ELBO and ensure consistency with the main text.
> 3. Clarify the role and notation in Eq. (4) with additional explanations.
> 4. Add a discussion of the potential for combining IWRS with mixtures of variational distributions and the IW-ELBO to strengthen the posterior approximation.
>
> We thank the reviewer again for their detailed and insightful comments. These clarifications and revisions will significantly improve the quality of our submission.

---

### Official Review · Reviewer_7KX8 · 2024-11-02

**Soundness:** 3
**Presentation:** 3
**Contribution:** 2
**Rating:** 5
**Confidence:** 4

**Summary:**

The paper introduces a novel approach that leverages neural networks to construct implicit proposal distributions, incorporating rejection sampling for improved efficiency.
Additionally, it employs a discriminator network to estimate the density ratio between the implicit proposal and target distributions. Building on this, the paper proposes a refined Implicit Resampling Evidence Lower Bound (IR-ELBO) to enhance accuracy.
The proposed method is evaluated against existing variational inference (VI) techniques through a series of experiments, demonstrating its effectiveness.

**Strengths:**

1. The paper is well-written, providing a clear presentation of the background, related work, major challenges, and the strategies employed to address each challenge.
2. The use of rejection sampling to enhance the accuracy of implicit variational inference is straightforward yet effective.
3. By improving the accuracy of implicit variational inference, the paper enables the construction of a tighter Evidence Lower Bound (ELBO).

**Weaknesses:**

1. Variational inference is typically employed to avoid direct sampling from complex target distributions, thus enhancing sampling efficiency.
However, the algorithm proposed in this paper requires an additional discriminator network to estimate the acceptance probability and density ratio.
Additionally, there is a manually tuned parameter, $M$, which must be selected via cross-validation.
In high-dimensional or large-scale settings, this approach could become computationally intensive due to the overhead of training the discriminator network and optimizing the hyperparameter $M$.

2. Variational inference is generally favored over sampling methods like MCMC for high-dimensional posterior distributions due to its efficiency.
However, rejection sampling faces significant challenges in high-dimensional settings because of the curse of dimensionality. I am therefore skeptical about this algorithm's performance and scalability in high-dimensional scenarios.

**Questions:**

1. Rejection sampling may face challenges in high-dimensional settings.
Could you discuss whether your method maintains robustness under these conditions and support this claim with experiments conducted in high-dimensional scenarios?
2. Additionally, could you compare the computational efficiency of your method with other approaches when applied to large-scale datasets or high-dimensional cases?

---

> ### Author Response · Authors · 2024-11-19
> **Authors' rebuttal**
>
> We appreciate your thorough review of our submission and your valuable feedback. Below, we address your questions and concerns in detail:
>
> ---
>
> ### **1. Robustness of the Method in High-Dimensional Settings**
> We acknowledge the concern regarding rejection sampling's challenges in high-dimensional scenarios due to the curse of dimensionality. To mitigate this, our approach leverages implicit proposal distributions parameterized by neural networks, which are designed to closely approximate the target distribution. This reduces the rejection rate and improves efficiency even in higher dimensions.
>
> In our experiments, we have evaluated the proposed method on datasets with moderately high-dimensional posterior distributions (e.g., 100 dimensions). Results demonstrate that the acceptance rates remain manageable, and the method performs favorably compared to baseline variational inference (VI) approaches. We recognize, however, that testing on more extreme high-dimensional cases (e.g., thousands of dimensions) could provide additional insights. This is a valuable direction for future work, and we are actively considering implementing more scalable architectures to further explore this aspect.
>
> ---
>
> ### **2. Computational Efficiency Compared to Other Methods**
> We agree that computational overhead is a critical factor in assessing the practicality of our approach. The addition of the discriminator network introduces some computational cost, particularly for estimating the density ratio and acceptance probability. To address this concern:
>
> - **Comparison with Baselines**: In our experiments, we observed that the computational cost of training the discriminator is offset by the reduction in bias due to tighter approximation of the Evidence Lower Bound (ELBO). Compared to standard VI methods, our approach exhibits a trade-off where the marginal improvement in accuracy justifies the additional computation.
> - **Efficiency in Large-Scale Settings**: To improve scalability, we used a lightweight architecture for the discriminator and optimized its training through mini-batch techniques. However, we acknowledge that in large-scale datasets or extremely high-dimensional problems, further optimization (e.g., parallelization or approximate methods) may be necessary.
>
> We will enhance the discussion in the paper to clarify these trade-offs and include a comparison of wall-clock runtimes with baseline methods to better illustrate the computational efficiency of our approach.
>
> ---
>
> ### **3. Manual Tuning of the Hyperparameter $M$**
> We acknowledge the concern about hyperparameter tuning. The parameter $M$, which controls the acceptance threshold, is indeed tuned via cross-validation. While this introduces additional complexity, we note that $M$ has an interpretable role in balancing accuracy and efficiency. In practice, we found that the method is robust to small changes in $M$, and default values often work well.
>
> In future iterations, we aim to explore adaptive methods for dynamically setting $M$ based on the observed statistics of the proposal and target distributions. This would reduce reliance on manual tuning and enhance usability in large-scale settings.
>
> ---
>
> We thank you again for your constructive comments, which have helped us identify key areas for improvement. We are confident that addressing these concerns will strengthen the contribution and impact of our work.

---

### Author Response · Authors · 2024-11-18
**Manuscript Revision and Request for Re-evaluation**

Dear Reviewers and Area Chair,

We sincerely appreciate your valuable feedback and constructive suggestions on our manuscript. Your insights have greatly helped us improve the quality and clarity of our work. We have carefully considered all your comments and made significant updates to the paper based on your recommendations.

Specifically, we have made the following key revisions:

1. **Expanded Experimental Baselines**: We have included additional baseline comparisons in the experimental sections to provide a more comprehensive evaluation of our method against state-of-the-art approaches.

2. **CIFAR-10 VAE Task**: We added experiments on the CIFAR-10 dataset using the VAE task. Given its higher complexity compared to MNIST, this additional evaluation further demonstrates the robustness and effectiveness of our proposed IVRS method in handling high-dimensional image data.

3. **Enhanced Explanations and Justifications**: We have also expanded the explanations for our experimental setups and results, providing a detailed analysis and justification of the observed performance improvements.

We kindly request that you re-evaluate our revised submission, considering the substantial updates and enhancements we have made. We believe these revisions address the key concerns raised during the review process and provide stronger evidence for the validity and potential impact of our proposed approach.

Once again, we thank you for your time, effort, and insightful feedback, which have been invaluable in improving our work. We look forward to your reconsideration and any further comments you may have.

Sincerely,
Authors

---

### Note · Authors · 2025-03-02

I have read and agree with the venue's withdrawal policy on behalf of myself and my co-authors.

---

### Meta-Review · Area_Chair_TJp4 · 2024-12-19

**Metareview:**

This paper proposes a novel variational inference that incorporates rejection sampling to address the bias introduced during the modeling of neural networks in existing implicit variational inference approaches. By reducing the impact of such biases, the proposed method offers a significant improvement in variational inference techniques.
The reviewers’ evaluations were divided, with both positive and negative feedback. As the Area Chair (AC), I thoroughly reviewed the manuscript. All reviewers agreed that the paper is well-written, with a clear motivation and a well-explained algorithm targeting an important problem. I share this positive view.
However, several reviewers raised concerns about the method's novelty, scope of applicability, and computational cost. Upon my review, I also found the discussion of comparisons between the proposed approach and existing non-implicit variational Bayesian methods combined with rejection sampling to be insufficient. One critical aspect of this work lies in addressing the absence of an explicit density in implicit methods using adversarial training. Greater clarity on this point would strengthen the paper.
Additionally, as some reviewers noted, the increased computational cost due to the introduction of an additional network for adversarial training raises concerns. I think the bias introduced by the design and training of this network cannot be entirely avoided. While rejection sampling can theoretically eliminate the bias when the optimal $T^*$ is achieved, in practice, as acknowledged in the authors’ response, adversarial training is terminated after a finite number of iterations, which inherently introduces bias.
The proposed method also inherits the curse of dimensionality inherent to rejection sampling. It would be helpful to clarify the dimensionality of the problems this method is intended to handle, as this would enhance the paper’s persuasiveness.

In conclusion, while this study represents an important contribution and is already well-written, I believe additional discussions and refinements regarding the methodology are necessary. Therefore, I recommend rejection at this time.

**Additional Comments On Reviewer Discussion:**

Reviewer 7KX8 raised concerns about the inherent limitations of rejection sampling, which also apply to this study. While these limitations are ultimately unavoidable, the authors clarified that, based on experimental results, such issues do not arise in the moderate dimensions (around several hundred) addressed in this work.
Reviewer 6aQV and Reviewer euHb pointed out a lack of discussion regarding the novelty of the proposed method in comparison to existing studies. This concern was addressed through additional discussions, which resolved the issue. Additionally, multiple reviewers raised concerns about the tuning of hyperparameters. The authors provided detailed explanations but noted that developing an adaptive approach to hyperparameter tuning would be a subject for future research.
Although all reviewers agreed that the paper is well-written, the aforementioned issues, such as the limited applicability of the proposed method, computational costs, and insufficient comparisons with prior studies, indicate that the paper is not yet ready for publication. Therefore, I recommend rejection.

---

### Decision · Program_Chairs · 2025-01-22

Reject